# Measuring the effects of a nurse-led intervention on frailty status of older people living in the community in Ethiopia: A protocol for a quasi-experimental study

Ayele Semachew Kasa[1,2]*, Peta Drury[1], Hui-Chen (Rita) Chang[1,3], Shu-Chun Lee[4], Victoria Traynor[1]

1 School of Nursing, Faculty of Science, Medicine, and Health, University of Wollongong, Wollongong (UOW), New South Wales, Australia, 2 Department of Adult Health Nursing, College of Medicine and Health Sciences, Bahir Dar University, Bahir Dar, Ethiopia, 3 School of Nursing and Midwifery, Western Sydney University, Parramatta South Campus, New South Wales, Australia, 4 School of Gerontology and Long-Term Care, College of Nursing, Taipei Medical University, Taipei, Taiwan

* ask255@uowmail.edu.au

**Data Availability Statement:** No datasets were generated or analysed during the current study. All

## Abstract

### Background

The recent recognition of the multidimensional features of frailty has emphasised the need for individualised multicomponent interventions. In the context of sub-Saharan Africa, few studies have examined: a) the frailty status of the older population; b) the level of frailty and its health implications and; c) the impact of a nurse-led intervention to reduce frailty.

### Objectives

This study aims to design, implement, and evaluate a nurse-led intervention to reduce frailty and associated health consequences among older people living in Ethiopia.

### Methods

The study will be conducted on 68 older persons using a pre-, post-, and follow-up single-group quasi-experimental design. Residents of Ethiopia, $\geq$60 years and living in the community will be invited to participate in a 24-week program designed to decrease frailty and associated health consequences. Data will be collected at three-time points: baseline, immediately after the intervention, and 12 weeks post-intervention. To determine the effect of the intervention, changes in frailty, nutritional status, activities of daily living, depression and quality of life scores will be measured. To measure the effect of a nurse-led intervention on the level of frailty among older people a generalised linear model (GLM) using repeated measures ANOVA will be used. Statistical significances will be set at p-values < 0.05.

### Discussion

The results of this study will determine the impact of a nurse-led intervention to reduce frailty amongst community-dwelling older people living in Ethiopia. The results of this study will

relevant data from this study will be made available upon study completion.

**Funding:** This work was supported by the Australian Government Research Training Program Scholarship in the form of University Postgraduate Award (UPA) as a student stipend. The funders had and will not have a role in study design, data collection and analysis, decision to publish, or preparation of the manuscript.

**Competing interests:** This study was funded in part by the Australian Government Research Training Program Scholarship in the form of University Postgraduate Award (UPA) as a student stipend. No additional external funding was received for this study.

**Abbreviations:** CHWs, Community Health Workers; HICs, High-Income Countries; ICMF, Integral Conceptual Model of Frailty; LICs, Low-Income Countries; LMICs, Lower Middle-income Countries; QOL, Quality of life; SSA, sub–Saharan Africa; TFI, Tilburg Frailty Indicator; UOW, University of Wollongong; WHO, World Health Organization.

inform the development of future interventions designed to reduce frailty in lower-income countries.

## Trial registration

The trial was registered in ClinicalTrials.gov with the identifier of NCT05754398.

## Background

In the upcoming decades, the ageing population will become a significant demographic phenomenon worldwide [1]. The elderly population in sub-Saharan Africa (SSA) is projected to double from approximately 34 million in 2005 to 67 million by 2030. SSA is experiencing a faster growth in the number of older individuals compared to developed countries, and this pattern is expected to persist [1–3]. Among SSA countries, Ethiopia ranks second in terms of the largest population of individuals aged 60 and above, following Nigeria [4]. The escalating challenges of malnutrition and non-communicable diseases, along with limited healthcare access and inadequate living conditions for older individuals in Ethiopia [5, 6], contribute to the state of frailty.

Frailty is a multidimensional geriatric syndrome that refers to a condition in which there is a decline in physical, cognitive, and psychological functioning [7, 8]. Frailty is often characterised by weakness, fatigue, unintended weight loss, slowed motor functioning, and decreased energy levels [9]. It can be caused by a combination of various factors, including age-related physiological changes, underlying health conditions, environmental factors, and lifestyle choices [10, 11]. Progressive declines in physical, mental, and social health are manifestations of frailty. These manifestations greatly affect older people's well-being and quality of life (QOL) [12]. Physical, mental, cognitive, and social domains of functioning change with age. Accumulation of problems in one or more of these domains of functioning are the features of frail people [13]. Studies on older people revealed that physical frailty varies significantly and is more prevalent when psychological frailty is considered [14]. Across all domains of frailty, mobility, nutrition, and cognitive function were identified as the most frequently identified factors of frailty [15].

As frailty is a multifactorial health problem, prior studies have emphasised the need for individualised and multifactorial interventions [16]. Numerous high-quality studies using interventional designs emphasised the importance of multifactorial frailty interventions in older people and developed different frailty interventions. Majority of the studies found that physical, nutritional, and cognitive interventional approaches were effective in reversing frailty among community-dwelling older people [17–19]. These studies have emphasised the need for further studies to evaluate the effectiveness of frailty interventions in other settings [20–23] (Table 1).

Few studies have examined the effect of a nurse-led intervention to reduce frailty amongst community-dwelling older people [17, 22, 29]. Prior studies investigating the effect of interventions designed to reduce frailty have predominantly led by general practitioners and physiotherapists [25, 32, 33]. As frontline healthcare providers, nurses have frequent and direct contact with older persons and are in a unique position to identify health promotion needs and provide education, counseling, and support to improve health outcomes [34]. Nurses are trained to care for older persons with complex conditions that requires a holistic approach [35]. Nurse-led interventions have the potential to enhance health outcomes and alleviate the burden on acute hospital services for frail older individuals residing in the community [36].

**Table 1. Interventional studies conducted to reduce frailty in older people.**

| Author, year, country | Design, Sample | Mean age | Interventions | Intervention led by | Outcomes | Conclusion/ recommendations |
|---|---|---|---|---|---|---|
| [24], Singapore | Non-RCT, 94 | 70.9 | • Eligible pre-frail participants were subsequently enrolled in a 4-month multi-disciplinary intervention programme comprising (i) once-weekly group-based exercise classes lasting 1 hour each session (total of 16 sessions) with individually prescribed home exercises for maintenance between sessions and (ii) group-based nutritional education (6 sessions).<br>• Group size was maintained at 8–10 participants to ensure that each participant received adequate attention. While the intensity of exercise was not measured, the target was to achieve at least moderate intensity as tolerated by the seniors.<br>• The exercises focused on strength, balance, and endurance training, with a warm-up and cool-down routine. | Multidisciplinary | • A significant improvement post-intervention was observed in lower limb strength and power.<br>• Reversibility of pre-frailty, and the benefit of multi-component intervention in improving physical performance of pre-frail older adults was observed. | • Future studies including longitudinal follow-up to examine the sustainability of improvements in physical performance beyond the immediate period post-intervention needed. |
| [25], Australia | RCT, 2016 | 83.3 | • Participants in the intervention group received a multi-factorial, interdisciplinary treatment program intended to target frailty for a 12-month period following randomization.<br>• The interventions were individually tailored to each participant based on their frailty characteristics as assessed at baseline. | Multidisciplinary (Physio, nurse,) | • There were no major differences between the groups with respect to secondary outcomes.<br>• The few adverse events that occurred were exercise-associated musculoskeletal symptoms. | • Future studies are needed to consider follow-up beyond the end of the intervention period. |
| [17], Singapore | RCT parallel group, 246 | 70 | • 90 minutes duration, on 2 days per week for 12 weeks followed by 12 weeks of home-based exercises.<br>• Participants performed the exercises in groups of 8 to 10.<br>• Each participant was provided with several supplements: iron and folate, vitamin B6, vitamin B12, calcium, and vitamin D taken daily for 24 weeks.<br>• In the first 12 weeks, participants attended 1 per week x 2-hour sessions of cognitive training.<br>• For the subsequent 12 weeks, participants attended 1 bi-monthly x 2 hours 'booster' sessions. | Nurse | • Frailty scores were reduced in all groups over 12 months.<br>• Compared with the control group, nutritional and cognition intervention were almost 3 times more likely of frailty reduction in the intervention group.<br>• Physical intervention was associated with 4 times higher odds of frailty reduction.<br>• Combination intervention was associated with the highest odds of frailty reduction. | • Intervene effectively to reduce level of frailty and possibly prevent future risks of hospitalization, functional dependency, institutionalization, and deaths |

*(Continued)*

**Table 1.** (Continued)

| Author, year, country | Design, Sample | Mean age | Interventions | Intervention led by | Outcomes | Conclusion/ recommendations |
|---|---|---|---|---|---|---|
| [26], New Zealand | RCT, 504 | 80.3 | • Participants were randomly allocated to receive an 8-week Senior Chef programme, a 10-week Steady As You Go programme, a 10-week combined and intervention (combined group), or a 10-week social programme (control group). | Multidisciplinary | • At the 24-month follow-up, there were no differences in mean Fried scores between the intervention groups and the control group. | • Ongoing meaningful support from local agencies might be required to ensure sustainability of the efficacy of such programmes. |
| [27], Austria | RCT, 80 | 83 | • A twice a weekly six strength exercise within a circuit training session and discussed nutrition-related aspects. The active control group performs cognitive training. | Trained nonprofessional volunteers | • The prevalence of frailty and impaired nutritional status decreased significantly over time.<br>• The prevalence of frailty decreased by 17% in the Physical training group and by 16% in the Social Support group. | • Home-based physical training, and nutritional, and social support intervention conducted by nonprofessionals is feasible and can help to tackle malnutrition and frailty in older people living at home. |
| [28] | RCT, 89 | NR | • The participants allocated to the 'Exercise and Nutrition' group participated in an exercise training and nutritional program (cooking class) once a week, and the 'Exercise' group participated in the exercise training program only. | Multidisciplinary | • The combined physical exercise training and nutritional intervention program has beneficial effects on several domains of HRQOL and handgrip strength in prefrail elderly women living in the community. | • Further studies are needed to examine approaches that facilitate maintenance of the improved outcomes by combined exercise training and nutritional intervention |
| [29], South Korea | Quasi-experimental pretest-posttest, 40 | 77.1 | • Session 1: Exercise and physical activity 2 times per week x 12 weeks<br>• Upper and lower limb resistance exercises.<br>• The routine was repeated for 30 minutes.<br>• Range of motion exercises as a cool-down exercise.<br>• Session 2: Nutritional and psychosocial interventions<br>• Nutrition education and counselling tailored to individuals' health status, chronic conditions, diet, household structure, and living environment.<br>• Education on mental health, such as depression relief and stress management.<br>• The intervention was provided once per week for approximately 20 minutes per session for 12 weeks.<br>• Data were collected using self-administered or face-to-face interviews three times: screening test, pre-test, and post-test. | Nurse | • Statistically significant improvements in frailty score, physical function and activity, nutritional status, and depression.<br>• No significant findings: ESSI. | • Long-term follow-up studies and additional statistical analyses needed. |

(*Continued*)

**Table 1.** (Continued)

| Author, year, country | Design, Sample | Mean age | Interventions | Intervention led by | Outcomes | Conclusion/ recommendations |
|---|---|---|---|---|---|---|
| [18], Spain | RCT, 172 | 78.3 | • Physical activity programme included two main components: (i) aerobic exercise consisting of walking outdoors for 30–45 min per day at least 4 days per week and (ii) a set of 15 mixed exercises (3 for strengthening arms, 7 for strengthening legs and 5 for balance and coordination) to be done at home for 20–25 min at least 4 days per week.<br>• Each exercise had to be repeated 10 times a minute (progressively increasing up to 15 times after 2–3 months), with a rest of half a minute between each set of exercises. | Multidisciplinary | • An intervention focused on physical exercise and maintaining good nutritional status may be effective in pre-venting frailty in community-dwelling pre-frail older individuals | • Further research is also needed regarding the main determinants of adherence to physical activity programmes for older people. |
| [22], South Korea | Quasi-Experimental study, 246 | 78.8 | • Multicomponent interventions.<br>• Exercise, cognitive training, and education on nutrition.<br>• 1 time per week x 12 weeks (Total 12 sessions).<br>• The researchers who collected and analysed the data were blinded to the participants' group allocation. | Nurse | • Non-significant reduction in frailty score from pre-intervention (T0) to 12-weeks follow-up (T2).<br>• Significant improvements in levels of depression, social activity, and social support | • Future studies are needed. |
| [30], UK | RCT, 83 | NR | • The intervention targeted at increasing independence and well-being through addressing key areas of mobility, nutrition, psychological well-being and social isolation. | Multidisciplinary | • A multicomponent health promotion intervention was acceptable and delivered at modest cost.<br>• The study shows promise for improving clinical outcomes, including functioning and independence. | • Future recommendations for a larger trial regarding intervention dosage. |
| [31], Netherlands | Quasi-Experimental study, 377 | 82 | • A multi-disciplinary meeting attended by at least the GP, the nurse practitioner, and a secondary-line geriatric nurse practitioner.<br>• Depending on frail elderly's problems discussed, the meeting was also attended by other health professionals such as geriatric physiotherapists, geriatricians, pharmacists, district nurse, nursing home doctors and mental health workers.<br>• The concrete actions, activities and responsibilities of these health professionals were discussed during this meeting. | GP-Led Multidisciplinary | • The Walcheren Integrated Care Model had a positive effect on love and friendship and a moderately positive effect on general quality of life.<br>• No significant differences were found on health outcomes such as experienced health, mental health, social functioning and functional abilities. | • Future research is needed to gain greater insight into what specific outcomes can be achieved. |

*(Continued)*

**Table 1.** (Continued)

| Author, year, country | Design, Sample | Mean age | Interventions | Intervention led by | Outcomes | Conclusion/ recommendations |
|---|---|---|---|---|---|---|
| [32], Australia | RCT, 241 | 83.2 | • Management of chronic health conditions and medication review.<br>• Participants were referred to services as indicated.<br>• Multiple strategies were used to maximise the amount of treatment received by participants allocated to the treatment group, such as involvement of family and carers, exercise diaries, visual cues, goal setting, and education. | Physiotherapy led Multidisciplinary | • Assuming a linear relationship between amount of treatment received and the effect of the treatment on frailty, the size of the effect on frailty is a reduction by 1.0 frailty criterion | Future studies are recommended. |
| [33], Netherlands | Quasi-Experimental study, 377 | 82 | • The general practitioner (GP) contacts the care recipient and informal caregiver to provide the opportunity for any last adjustment.<br>• A case manager implements the care plan and coordinates care delivery. Periodic evaluations of the care plan ensure adequate monitoring of the needs of the care recipient and the informal caregiver. | GP-Led and Multidisciplinary | • The intervention significantly contributed to the reduction of subjective burden and significantly contributed to the increased likelihood that informal caregivers assumed household tasks.<br>• No effects were observed on perceived, health, time investment and quality of life. | • Future interventions and research are provided |

ESSI: Enriched Social Support Instrument, NR: Not reported.

A recent study in Ethiopia has identified 39% of community-dwelling older people were living with frailty [37]. Frailty impacts quality of life of older people and places additional demands on the healthcare system [38]. The majority of frail older person within Ethiopia are living in the community, where healthcare predominantly delivered by community nurses. Therefore, the aim of this study is to design, implement, and evaluate the effects of a nurse-led intervention on the frailty and quality of life of older people living in the community in Ethiopia.

## Methods

### Study hypothesis and design

The primary outcome is change in frailty status of community-dwelling older persons measured at three points in time: baseline (T0), immediately post-intervention (T1) and at 12 weeks post intervention (T2) using the Tilburg Frailty Indicator-Amharic Version (TFI-AM) [39]. We hypothesise that frail older people who received the nurse-led intervention will have a reduced frailty score, including the physical, psychological, and social domains of frailty. Secondary outcomes include changes in the activities of daily living as measured using the Katz Index of Independence in Activities of Daily Living [40], nutritional status using Mini-Nutritional Assessment (MNA) [41–43], Recent appetite using the Simplified Nutritional Appetite Questionnaire (SNAQ), level of depression using Geriatric Depression rating Scale-15 (GDS-

15) [43–45] and quality of life using the World Health Organization's QOL Questionnaire (WHOQOL-BREF) [46] measured at each data collection time points (Table 2).

The study will be conducted using a pre-, post-, and follow-up single group quasi-experimental design. A quasi-experimental design was chosen since randomising clients at these community programs for vulnerable older people would not be ethical or feasible (33–35).

This study will adhere to the Transparent Reporting of Evaluations with Nonrandomised Designs (TREND) guidelines [63]. This protocol adheres to the Standard Protocol Items: Recommendations for Interventional Trials (SPIRIT) reporting checklist [64]. The protocol was registered retrospectively three months after data collection commenced. The only amendment requested from the clinical trial review was to provide additional detail on the secondary outcome measures. This detail was provided and the trial was registered at ClinicalTrial.gov. The trial is being undertaken exactly as described in NCT05754398.

## Setting

The study will be conducted in Bahir Dar, Ethiopia. Bahir Dar is the capital city of the regional state of Amhara, Ethiopia. Based on a survey conducted by Bahir Dar City Labour and Social Affairs Administration Office in 2018 revealed that there were over 3,300 older people in Bahir Dar City administration [65]. In Bahir Dar City, there are three public and four private hospitals, 10 health centers and 15 health posts. In the Ethiopian health tier system, health posts, health centres and primary hospitals are grouped under primary health care, whereas general hospitals and specialized hospitals are grouped as secondary and tertiary level healthcare respectively [66]. The health office of Bahir Dar city administration oversees all primary level healthcare facilities in the city administration [67].

## Sample size

The study sample size was calculated using a priori computation of sample size using G* Power version 3.1.9.4 [68] with assumption of a two-tailed test with an alpha value of 0.05, effect size (f) of 0.5, and a power of 0.95. The power was set using a Wilcoxon signed-rank test based on normal parent distribution methods. Using this equation, we estimate that 57 participants will be recruited for this study. By considering a 10 to 20% [23, 29] withdrawal rate during the intervention, at least 68 study participants will be required.

## Eligibility

**Inclusion and exclusion criteria.**   The year in which 'old age' commences is determined by a setting and the formal cutoff point legislated in social policy for each country [56]. In Ethiopia, the cutoff point for old age is 60 years [69, 70]. Therefore, older people 60 years or above, whose frailty score $\geq 5$ as measured by the Tilburg Frailty Indicator Amharic Version (TFI-AM) [39] and residing in Bahir Dar, Ethiopia, will be included in the study. Participants will be excluded if they are unable to communicate, have major cognitive impairment, are bed-redden, do not live at home, have been hospitalised with a known psychiatric problem within the past six months, or will not reside in the selected area during the study period.

**Recruitment.**   A list of older people in the selected sub city will be obtained from the household's registration which are listed with the city's administration health office. Then, potential study participants will be recruited by the Community Health Workers (CHWs) using a convenience sampling method in home-to-home bases. During the home visit the CHW will explain the aim of the study, undertake a screen to determine frailty status, and obtain consent to participate in the intervention.

**Table 2. Study outcome, measurement tool description, and measurement frequency.**

| Section | Outcome measures | Measurement tool description | Measurement frequency | | |
|---|---|---|---|---|---|
| | | | T0 | T1 | T2 |
| I | Participants' socio-demographic information. | Participants' socio-demographic characteristics and other factors. | ✓ | x | x |
| II | Health-related factors | Participant health-related information including medical history, health care service utilisation, living arrangement, lifestyles, or personal related behaviours such as Khat chewing, smoking and drinking habits. | ✓ | x | x |
| III | Change in frailty status of community-dwelling older people. | Changes in frailty will be measured using the Tilburg Frailty Indicator (TFI) [47–49]. The TFI comprises 15 self-reported questions, divided into three distinct domains. The physical, psychological, and social domains are the three distinct domains that constitute the TFI. The physical domain consists of eight questions related to different physical health of older people. The psychological domain contains four items related to the psychological health of older people. The last domain, the social domain has three questions related to social relations. Eleven items have responses scored as "yes" or "no", and four items have responses scored as "yes", "no," or "sometimes". The instrument's total score ranges from 0 to 15: higher scores indicate worse frailty. Frailty is diagnosed when the total TFI score is $\geq$5 [49, 50]. To create a suitable frailty measurement tool for Ethiopians older people, the original TFI instrument was cross-culturally adapted, and it is approved by the original author. | ✓ | ✓ | ✓ |
| IV | Change in the nutritional status of community-dwelling older people. | Changes in nutritional status of community-dwelling older people will be measured using the Mini-Nutritional Assessment (MNA) tool. MNA is widely used including in Ethiopia [41–43] and was developed specifically for use in older people [43, 51]. The tool appears to be the most appropriate nutrition screening tool for use in community-dwelling older people [52]. The MNA score ranges from 0 to 30. A higher score indicates an improved nutritional status. Moreover, based on the MNA score, the nutritional status of older people will be classified as: Malnourished (MNA Score <17), at risk of malnutrition (MNA Score 17 to 23.5), or normal nutritional status (MNA Score 24 to 30) [42]. | ✓ | ✓ | ✓ |
| VI | Recent appetite | Recent appetite status of the older people will be measured using the Simplified Nutritional Appetite Questionnaire (SNAQ). The SNAQ is a brief, valid, and reliable four-item survey tool with a maximum score of 20 points and a score of <14 points, indicate significant risk of at least 5% weight loss within six months [53, 54]. | ✓ | ✓ | ✓ |
| V | Change in the level of depression of community-dwelling older people | Changes in depression status of community-dwelling older people will be measured using the Geriatric Depression rating Scale-15 (GDS-15). The scale has been extensively tested and validated in low and middle-income countries and across cultures [55] and being used in Ethiopia [43–45]. The values range from 0 to 15: higher scores indicate worse depression. Depression is considered present using a cut-off point $\geq$5. Scores of 0 to 4 indicate no depression, 5 to 8 indicates mild depression, 9 to 11 indicates moderate depression and 12 to 15 indicates severe depression [44, 56]. | ✓ | ✓ | ✓ |
| VII | Social support | Study participants' social support will be measured with the short version of the Social Support Questionnaire (SSQ-6). The SSQ-6 is a six-item instrument used to determine the number of people involved in providing support and to measure the level of satisfaction with the support. The satisfaction items are to be rated on a 6-point Likert scale, with the possible total score of 6–36; a higher score indicating more satisfaction with the available social support. This tool was found to have strong internal consistency and showed satisfactory validity [57]. | ✓ | x | x |
| VIII | Changes in the activities of daily living | Activities of daily living will be measured using the Katz Index of Independence in Activities of Daily Living [40]. The Index ranks adequacy of performance in the six functions of bathing, dressing, toileting, transferring, continence, and feeding [58]. The responses are scored as a 'Yes' or 'No' for independence in each of the six functions. The instrument's total score ranges from 0 to 6: higher scores indicate better activities in daily living. A score of 2 or less indicates severe functional impairment, 4 indicates moderate impairment, and 6 indicates full function in activities of daily living [59, 60]. | ✓ | ✓ | ✓ |
| IX | Change in quality of life (QOL) of community-dwelling older people. | World Health Organization's QOL Questionnaire (WHOQOL-BREF) will be used to measure the changes in quality of life of older people. WHOQOL-BREF is a self-report questionnaire or interviewer-administered which consists of 26 questions categorized into four domains that are scored on a 5-point Likert scale. The four domains are physical health (7 items), psychological health (6 items), social relationships (3 items), and environment (8 items) [56, 61, 62]. Values will be transformed into scores ranging from 0 to 100 according to the WHO guidelines. Higher scores indicate higher the quality of life [46]. Two (question number 1 and 2) items measure the overall QOL and general health. | ✓ | ✓ | ✓ |

MNA: Mini-Nutritional Assessment, QOL: Quality of life, SNAQ: Simplified Nutritional Appetite Questionnaire, TFI: Tilburg Frailty Indicator,

Each participant will undergo a baseline assessment before starting the nurse-led intervention after confirming eligibility, willingness, and receiving written informed consent. The baseline assessment will include socio-demographics, health-related factors, frailty, nutrition, depression, social support, activities of daily living, and quality of life. At the end, study participants who received all the nurse-led intervention sessions will be included in the final analysis to determine the effectiveness of the nurse-led intervention to reduce frailty among older people in Ethiopia.

**Authors access to study participant information and confidentiality.** All information related to study participants will remain confidential and will be identifiable by codes known only to the researcher. Study participants' involvement in the study is entirely voluntary and participants may choose to withdraw at any time.

## Intervention

The design of the intervention has been guided by the Integral Conceptual Model of Frailty (ICMF) framework. This framework denotes that physical, psychological, and social domains of health are key components to ensure the health of frail older people [13, 71].

The Nurse-led Intervention (NLI) program comprises six distinct and interconnected education sessions including:

- Session 1 Ageing and age-related changes

- Session 2 Healthy nutrition

- Session 3 Physical activity

- Session 4 Mental health

- Session 5 Social interaction and support

- Session 6 Discussion and reflection (Fig 1):

The CHWs will deliver one face-to-face session per month to each participant in their home. Therefore, all six sessions will take six consecutive months to deliver. Each session will last approximately from 30 to 40 minutes. During the six months when the intervention is delivered, there will be a fortnightly 5 to 10-minute follow-up phone call with participants to receive feedback about the education sessions and provide opportunistic counseling on the specific topics. Each CHW will be trained by a PhD candidate in nursing. To reduce loss to follow-up (LTFU) and increase adherence rates to intervention, participants will be encouraged and reminded by phone to attend upcoming sessions (Table 3).

To promote adherence to the intervention, a nurse-led education intervention handbook contextually relevant to frailty management for older people will be developed. The content of the training handbook will be based on the multi-dimensional concept of frailty [15, 22, 72] and will be customised to the local setting. The training handbook will be accompanied by illustrative pictures. The training handbook will be reviewed by Ethiopian community nurses with experience in community health care services. The training handbook will be translated into the local language, Amharic, and reviewed by a bilingual expert from Bahir Dar University, Ethiopia. A booklet on frailty management education will be disseminated to the study participants during the first session of the nurse-led education. To promote ease of access to education sessions, the education sessions will be conducted at the participants' own home. Each CHW will be provided with a notebook to record the progress of each participant undertaking the program and any questions that need to be followed up at a subsequent session.

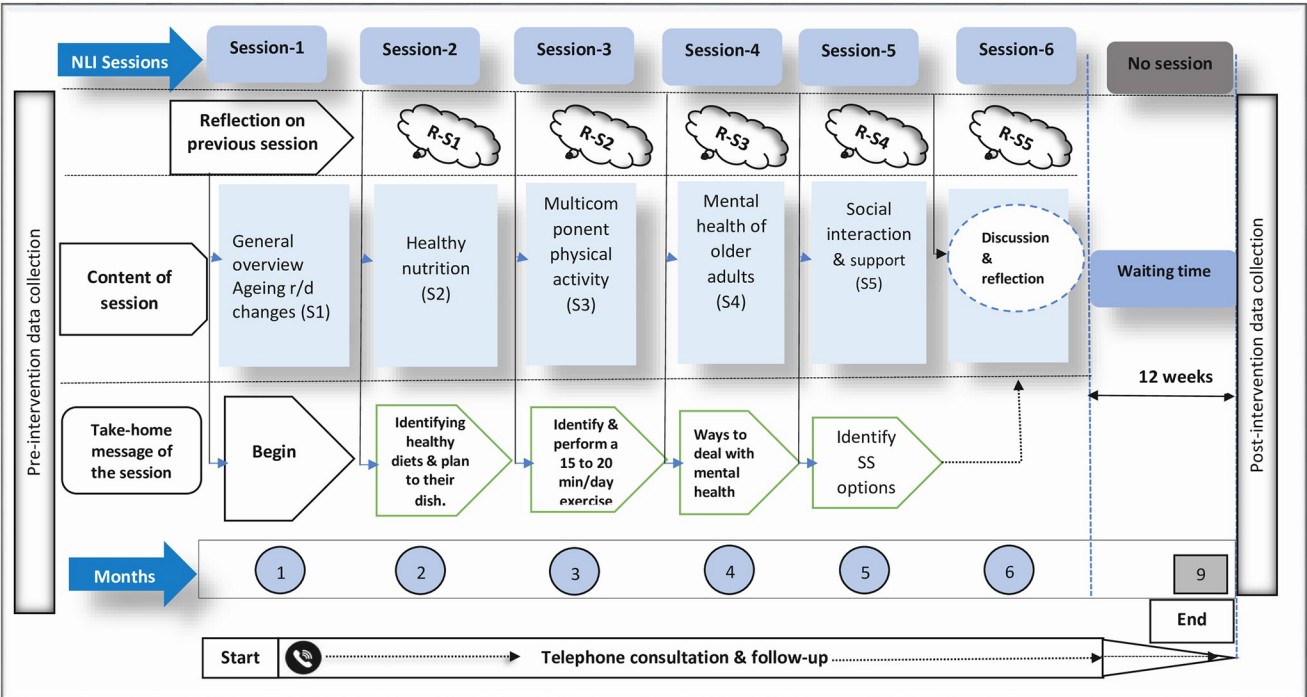

**Fig 1. Construct of the nurse-led intervention (NLI) program.** R-S1: Reflection on session 1, R-S2: Reflection on session 2, R-S3: Reflection on session 3, R-S4: Reflection on session 4, R-S5: Overall discussion on all sessions, R-S6: Reflection on session SS: Social support.

**Recruiting process of community health workers (CHWs) to deliver the intervention.**
Discussion on the overall aims and procedures of the study was done with senior health facility administrators in the study area. Two senior health administrators from the zonal and health centre in Bahir Dar were contacted to discuss the details of the study objectives, and procedures to be followed. Further discussion made on recruiting potential CHWs that will be appropriate to deliver the intervention who have previous experience providing health promotional interventions. Moreover, when selecting the potential CHWs, the discussion also focused on communication, commitment to work, compassion, and professionalism.

Community Health Workers are registered nurses employed by the local government and work closely with the local community home-to-home and at health posts [73]. CHWs know the culture, lifestyle, and social norms of the community and provide culturally appropriate health education and information, help community members access the care they need, counsel and guide on health-promoting behaviours, and for the health needs of individuals and communities [74].

Before the intervention is initiated, two CHWs undergo training by the PhD candidate in nursing on the study aim, and procedure, exercise safety protocols, and how study participants be approached ethically. Moreover, an observational checklist has been used to assess the communication skill, compassion, and knowledge of the CHWs on the contents of the intervention. The intervention checklist has been prepared based on the contents of the six sessions of the intervention handbook.

## Data collection

To reduce assessor bias, CHWs will not be involved in the data collection process. Two registered nurses from Bahir Dar city will be recruited for data collection. They will be required to

**Table 3. Intervention description.**

| | Intervention description | | |
|---|---|---|---|
| | **Session Focus** | **Content** | **Description** |
| Session 1 | Age and age-related changes contribute to frailty in older people. | • Introduction to ageing and age-related changes.<br>• General changes and symptoms that happen in older people.<br>• Sensory and perceptual changes that may happen in older people.<br>• Other changes that may happen in later life. | CHWs will:<br>• Introduce the contents of the session.<br>• Ask leading questions related to the session.<br>• Deliver and discuss the session.<br>• Give a take-home message and set an appointment for session 2. |
| Session 2 | Healthy nutrition for older people. | • The need for a healthy diet (Calcium, Vitamin D, Vitamin C, Omega-3, Protein, Fiber, cereals, fruit, and vegetables)<br>• Factors that cause nutritional problems in older people. | The CHWs will:<br>• Give the opportunity to the study participant to reflect on their take-home messages given at session 1.<br>• Ask leading questions related to session 2.<br>• Deliver and discuss session 2.<br> • Sources of a healthy diet in the local setting.<br> • Daily feeding habits, frequency, and composition of diet.<br> • Selected consumption markers for protein intake.<br> • Fruit and vegetables needed in a dish.<br> • Fluid intake habits.<br>• give a take-home message and appointment for session 3. |
| Session 3 | Physical activity for older people. | • Introduction of the session.<br>• Benefits of physical exercise.<br>• On how older people initiate exercising safely.<br>• Frequency of physical exercise.<br>• Types of exercises:<br>i. Cardiovascular fitness activities:<br> • Brisk walking, swimming, cycling, tennis, and household chores.<br>ii. Strength activities:<br> • Weightlifting, resistance, and climbing stairs.<br>iii. Flexibility activities:<br> • Stretching, mopping, and gardening<br>iv. Balancing activities:<br> • Heel raise, half squat, and side leg raise exercises. | The CHWs will:<br>• Give the opportunity to the study participant to reflect on their take-home messages given at session 2.<br>• Introduce the content of session 3.<br>• ask leading questions related to session 3.<br>• Deliver and discuss session 3.<br> • For cardiovascular fitness activities, study participants will be encouraged to select one activity and perform it for 2 to 5 minutes resting 1 minute between for 15 minutes twice a day.<br> • For strength activities, study participants will be encouraged to select one activity and perform 3 sets of 12 repetitions and rest 1 minute between sets about 2 to 3 times a week.<br> • For flexibility activities, study participants will be encouraged to select one activity and perform 3 sets of 3 repetitions and rest for 30 seconds between sets every day.<br> • For flexibility activities, study participants will be encouraged to select one activity and perform for 3 sets for 2 to 5 minutes resting 1 minute between sets 2 to 3 times a week.<br>• give a take-home message and appointment for session 4. |
| Session 4 | Mental Health in older people | • Overview of mental health.<br>• Contributing factors to mental health.<br>• Strategies to improve mental well-being. | The CHWs will:<br>• Give the opportunity to the study participant to reflect on their take-home messages given at session 3.<br>• Introduce the content of session 4.<br>• Ask leading questions related to session 4.<br>• Deliver and discuss session 4.<br>• Give a take-home message to the study participant and appointment for session 5. |
| Session 5 | Social interaction and support for older people. | • The importance of social connections.<br>• Social isolation of older people.<br>• The risk of social isolation. | The CHWs will:<br>• Give the opportunity to the study participant to reflect on their take-home messages given at session 4.<br>• Introduce the content of session 5.<br>• Ask leading questions related to session 5.<br>• Deliver and discuss session 5.<br>• Give a take-home message and appointment for session 6. |

(*Continued*)

**Table 3.** (Continued)

| | Intervention description | | |
|---|---|---|---|
| | **Session Focus** | **Content** | **Description** |
| Session 6 | Overall discussion on the sessions | • Any age-related changes experienced by the participant.<br>• The importance and health benefits of a healthy diet.<br>• The importance of engaging in regular physical activity under their scope.<br>• The importance of building a great social network with neighbours and significant others.<br>• Strategies to sustain the intervention in the day-to-day activity in future life. | The CHWs will:<br>• Give the opportunity to the study participant to reflect on their take-home messages given at session 5.<br>• Introduce the content of session 6.<br>• Ask leading questions related to session 6.<br>• Deliver and discuss session 6.<br>• Encourage the study participant to plan on how to contact the community health workers to receive further support and counselling to sustain the interventions. |

CHWS: Community Health Workers

attend a 2-day training workshop focused on data collection tools, and ethical consideration for the older person. Data will be collected in person through a face-to-face administered questionnaire. The following physical data will also be collected at the same time: height, weight, calf circumference and mid arm circumference. These parameters will assist in determining nutritional status in conjunction with the other items in the Mini-Nutritional Assessment (MNA) tool [41, 51, 75] Questionnaires will be administered to study participants at baseline (before the intervention) (T0), immediately after the intervention (T1) and at the 12 weeks of post intervention (T2) [29, 76, 77]. The structured questionnaire comprises nine different sections of which seven sections (Section III to IX) have been developed from validated tools (Table 2).

To measure adherence to the program the CHWs will maintain an attendance record. At the beginning of each session the CHWs will record attendance of the participants by checking off the name of a participant against each session in their notebook. Acceptability of the program from the perspectives of the CHWs will be recorded in two ways: (1) field notes written by the CHWs in the notebooks they used to while delivering the program and (2) at the end of the intervention. The informal discussions with the CHWs will also be used to generate feedback about the training handbook. Acceptability of the program from the perspectives of the participants will also be recorded at the end of the last session of the intervention.

## Data analyses

The information collected through paper-based questionnaires will be inputted into EpiData Manager software and then exported to IBM SPSS 26.0 (IBM Corp., Armonk, NY, USA) for analysis. The correlation between frailty and various factors among older individuals will be assessed using Pearson correlation analysis. Numerical data will be summarised as mean (±SD), while categorical data will be presented and summarised using both frequencies and percentages. The normality of the data will be checked using a One-Sample Kolmogorov-Smirnov's test. To evaluate the impact of the nurse-led intervention, a generalized linear model (GLM) with repeated measures ANOVA will be employed. If the data do not follow a normal distribution, the Friedman test will be utilised. Student's t-test will be used to compare the level of frailty with categorical variables. Statistical significance will be determined using a p-value threshold of $< 0.05$.

## Discussion

Frailty in older people is influenced by a wide range of physical, behavioural, and psychosocial health factors. The recent recognition of the multi-dimensional nature of frailty has highlighted the need for individualised multi-factorial interventions. Studies from developed countries have recognised the importance of frailty in older people and have developed several frailty interventions with positive outcomes from community settings. The proposed study is aimed to examine the effect of nurse-led intervention on frailty status among older people living in a low-income setting.

No prior studies have attempted to examine the effectiveness of nurse-led intervention designed to reduce the frailty status of older people in African. In resource-limited settings, there is limited focus on the healthcare of older people. This study aims to develop, implement and evaluate an intervention to reduce frailty in lower income settings. By conducting this research, we will contribute valuable knowledge to future researchers designing interventions to reduce frailty. Healthcare professionals, especially nurses who work in the community, will become familiar with screening tools for frailty. They will also have access to the interventional handbook developed by the researchers highlighting the six-month intervention and how to deliver it in the community.

This study will open the door for researchers and concerned government officials to consider the multidimensional healthcare needs of older people living in resource-limited settings.

Having a better understanding of older people who live in resource-limited settings will allow healthcare professionals, researchers, and government officials to consider the multidimensional healthcare needs of these group of population. The findings from this study will also contribute to frailty management strategies in reducing the adverse outcomes related to frailty and assisting in clinical decision-making.

### Strengths of the study

- This study is the first nurse-led intervention designed to decrease frailty among older people in Ethiopia.

- The study will demonstrate alternatives on how the nurse-led intervention for older people with frailty can be integrated with the existing Ethiopian health extension package.

- Healthcare professionals, working in the community through home visiting will become familiar with the concept of frailty and associated health outcomes when assessing older people.

### Limitations of the study

- At the start of each session the CHWs will ask the participants about their progress since the previous session, that is how they are incorporating the health information provided into their daily lives and will answer any questions from the participants about the previous session. However, no data collection tool was created to specifically record compliance with the health information provided in each of the sessions.

- The study will be time-consuming and with a relatively long follow-up, loss to follow-up may be an issue.

## Supporting information

**S1 Checklist. SPIRIT 2013 checklist: Recommended items to address in a clinical trial protocol and related documents\*.**
(DOC)

**S2 Checklist.** *PLOS ONE* **clinical studies checklist.**
(DOCX)

**S1 File. Inclusivity in global research questionnaire ASK.**
(DOCX)

**S2 File. Original study protocol Ayele.**
(DOCX)

## Author Contributions

**Conceptualization:** Ayele Semachew Kasa, Peta Drury, Hui-Chen (Rita) Chang, Shu-Chun Lee, Victoria Traynor.

**Methodology:** Ayele Semachew Kasa, Peta Drury, Hui-Chen (Rita) Chang, Shu-Chun Lee, Victoria Traynor.

**Writing – original draft:** Ayele Semachew Kasa, Peta Drury, Hui-Chen (Rita) Chang, Shu-Chun Lee, Victoria Traynor.

**Writing – review & editing:** Ayele Semachew Kasa, Peta Drury, Hui-Chen (Rita) Chang, Shu-Chun Lee, Victoria Traynor.

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
