## [Decision Letter · Decision Letter 0]

30 Aug 2023

PONE-D-23-00901Measuring the Effects of a Nurse-led Intervention on Frailty Status of Older People Living in the Community in Ethiopia: A Protocol for a Quasi-experimental StudyPLOS ONE

Dear Dr. Kasa,

Thank you for submitting your manuscript to PLOS ONE. After careful consideration, we feel that it has merit but does not fully meet PLOS ONE’s publication criteria as it currently stands. Therefore, we invite you to submit a revised version of the manuscript that addresses the points raised during the review process.

ACADEMIC EDITOR: Please submit your revised manuscript by Sep 18 2023 11:59PM. If you will need more time than this to complete your revisions, please reply to this message or contact the journal office at plosone@plos.org. Please include the following items when submitting your revised manuscript:A rebuttal letter that responds to each point raised by the academic editor and reviewer(s). You should upload this letter as a separate file labeled 'Response to Reviewers'.A marked-up copy of your manuscript that highlights changes made to the original version. You should upload this as a separate file labeled 'Revised Manuscript with Track Changes'.An unmarked version of your revised paper without tracked changes. You should upload this as a separate file labeled 'Manuscript'.

We look forward to receiving your revised manuscript.

Kind regards,

Azmeraw Ambachew Kebede, MSc

Academic Editor

PLOS ONE

Journal Requirements:

2. Please include the following request for minor text overlap and do not ping with follow up:

"We noticed you have some minor occurrence of overlapping text with the following previous publication(s), which needs to be addressed:

- https://doi.org/10.3390/ijerph17186660

In your revision ensure you cite all your sources (including your own works), and quote or rephrase any duplicated text outside the methods section. Further consideration is dependent on these concerns being addressed."

3. Please include the following request in the decision letter, and ping me with follow up." Please include a complete copy of PLOS’ questionnaire on inclusivity in global research in your revised manuscript. Our policy for research in this area aims to improve transparency in the reporting of research performed outside of researchers’ own country or community. The policy applies to researchers who have travelled to a different country to conduct research, research with Indigenous populations or their lands, and research on cultural artefacts. The questionnaire can also be requested at the journal’s discretion for any other submissions, even if these conditions are not met.  Please find more information on the policy and a link to download a blank copy of the questionnaire here: https://journals.plos.org/plosone/s/best-practices-in-research-reporting. Please upload a completed version of your questionnaire as Supporting Information when you resubmit your manuscript.

4. Please ask the authors to include an explanation for the retrospective CT registration and confirmation that all related CTs are registered, using send back in ITC desk notes. At RTC, please check the authors' response and ping me if the authors do not address this.

"The funders had and will not have a role in study design, data collection and analysis, decision to publish, or preparation of the manuscript."

7. We note that you have stated that you will provide repository information for your data at acceptance. Should your manuscript be accepted for publication, we will hold it until you provide the relevant accession numbers or DOIs necessary to access your data. If you wish to make changes to your Data Availability statement, please describe these changes in your cover letter and we will update your Data Availability statement to reflect the information you provide.

Reviewers' comments:

Reviewer's Responses to Questions

**Comments to the Author**

1. Does the manuscript provide a valid rationale for the proposed study, with clearly identified and justified research questions?

Reviewer #1: Yes

Reviewer #2: Yes

2. Is the protocol technically sound and planned in a manner that will lead to a meaningful outcome and allow testing the stated hypotheses?

Reviewer #1: Partly

Reviewer #2: Yes

3. Is the methodology feasible and described in sufficient detail to allow the work to be replicable?

Reviewer #1: Yes

Reviewer #2: Yes

4. Have the authors described where all data underlying the findings will be made available when the study is complete?

Reviewer #1: Yes

Reviewer #2: No

5. Is the manuscript presented in an intelligible fashion and written in standard English?

Reviewer #1: Yes

Reviewer #2: Yes

6. Review Comments to the Author

You may also provide optional suggestions and comments to authors that they might find helpful in planning their study.

Reviewer #1: You have clearly identified the rationale for the study. Developing an education package and delivering it is an ambitious project. I wish you well with this.

As a new program, I would think determining its acceptability and identifying any areas for modification are important outcomes. Yet I see no mention of acceptability/ adherence measures as outcomes.

How will the program’s acceptability with those delivering it and those reviewing it be assessed? Will you seek feedback on the training book? Will you measure adherence? If so, how? Meeting with the CHWs each month would be one measure. How do you determine compliance with the recommendations for exercise or diet? Or is this not possible?

Methods: Line 103

Please include the name of the instruments used to measure the primary outcome (frailty status) and the instruments used for the secondary measures. Table 3 lists and outlines the outcome measures (i.e. the score range, number of questions and a reference for the instrument). Maybe refer to this table earlier in the text in the methods, rather than later (in the data collection), or move text from this table into the methods section.

EDITS:

Line 123: as your sample calculation was an a priori analysis, change from future tense to past tense, i.e. sample size WILL be calculated, to sample size WAS calculated ….

Line 154 – change singular to plural, i.e. study participant’s to study participants’ involvement.

Reviewer #2: This single group study aims to design, implement, and evaluate a nurse-led intervention to reduce frailty among older people in Ethiopia. Data will be repeatedly collected, at baseline, immediately after intervention, and 12 weeks post-intervention. Data will be collected on frailty, nutrition status, activities of daily living, depression and quality of life.

Minor revisions:

1- Abstract: Grammatical error: Statistical

2- Line 126: Indicate the statistical testing method which achieves 95% power.

3- Line 220: Categorical data is typically summarized using both frequencies and percentages.

4- Line 220: State the statistical approach that will be used to test for normality of the data.

5- Since data will be collected at 3 time points, specifically indicate which outcomes and time points will be compared using Fisher’s exact tests, paired sample t-tests, and GLM with repeated measures ANVOVAs.

7. PLOS authors have the option to publish the peer review history of their article (what does this mean?). If published, this will include your full peer review and any attached files.

Reviewer #1: No

Reviewer #2: No

---

## [Author Response · Author response to Decision Letter 0]

11 Oct 2023

Dear Editor, we would like to thank you for the time to review our manuscript and for sharing supporting reference materials. In the current version of the manuscript, we included a figure that reflects the construct of the intervention. All the amendments and revisions made are highlighted in light yellow colour.

Academic editor’s comment: Please ensure that your manuscript meets PLOS ONE's style requirements, including those for file naming. The PLOS ONE style templates can be found at

Author’s response/explanation: Thank you for sharing the link to access the templates. We assure you that the manuscript meets PLOS ONE’S style requirements, including file naming. In the revised manuscript a figure has been included (see p. 7, line 170). 

Academic editor’s comment: Please include the following request for minor text overlap and do not ping with follow up: "We noticed you have some minor occurrence of overlapping text with the following previous publication(s), which needs to be addressed:- https://doi.org/10.3390/ijerph17186660. In your revision ensure you cite all your sources (including your own works), and quote or rephrase any duplicated text outside the methods section. Further consideration is dependent on these concerns being addressed."

Author’s response/explanation: For the requested minor textual overlaps, we ensured all the sources were cited.

We thought that we acknowledged the sources of information with appropriate citations. For example, information taken from the previously mentioned source/publication was cited as indicated using reference number 24. In the meantime, if still there are some concerns, we kindly wait for your advice. 

Academic editor’s comment: 3. Please include the following request in the decision letter, and ping me with follow up." Please include a complete copy of PLOS’ questionnaire on inclusivity in global research in your revised manuscript. Our policy for research in this area aims to improve transparency in the reporting of research performed outside of researchers’ own country or community. The policy applies to researchers who have travelled to a different country to conduct research, research with Indigenous populations or their lands, and research on cultural artefacts. The questionnaire can also be requested at the journal’s discretion for any other submissions, even if these conditions are not met. Please find more information on the policy and a link to download a blank copy of the questionnaire here: https://journals.plos.org/plosone/s/best-practices-in-research-reporting. Please upload a completed version of your questionnaire as Supporting Information when you resubmit your manuscript.

Author’s response/explanation: The requested PLOS’ questionnaire on inclusivity in global research was completed and uploaded as ‘Supporting Information’. 

Academic editor’s comment: Please ask the authors to include an explanation for the retrospective CT registration and confirmation that all related CTs are registered, using send back in ITC desk notes. At RTC, please check the authors' response and ping me if the authors do not address this.

Author’s response/explanation: Update made to detail about clinical trial registration to ensure clarity about when the clinical trial was registered (see p.5, lines 115-119).

Academic editor’s comment: Thank you for stating the following financial disclosure: "The funders had and will not have a role in study design, data collection and analysis, decision to publish, or preparation of the manuscript."At this time, please address the following queries:

Author’s response/explanation: Regarding the financial disclosure, we want to assure you the following:

a. The source of funding for this study has been stated in the ‘funding statement’ section (see p.12, lines 284 to 287)

b. The funders had no role in study design, data collection, and analysis, decision to publish, or preparation of the manuscript.

c. Not applicable. 

d. The authors received no specific funding for this work except the student received a support Scholarship in the form of the University Postgraduate Award (UPA). 

Amendments are included in the cover letter and indicated.

Academic editor’s comment: 6. In your Data Availability statement, you have not specified where the minimal data set underlying the results described in your manuscript can be found. PLOS defines a study's minimal data set as the underlying data used to reach the conclusions drawn in the manuscript and any additional data required to replicate the reported study findings in their entirety. All PLOS journals require that the minimal data set be made fully available. For more information about our data policy, please see http://journals.plos.org/plosone/s/data-availability. Upon re-submitting your revised manuscript, please upload your study’s minimal underlying data set as either Supporting Information files or to a stable, public repository and include the relevant URLs, DOIs, or accession numbers within your revised cover letter. For a list of acceptable repositories, please see http://journals.plos.org/plosone/s/data-availability#loc-recommended-repositories. Any potentially identifying patient information must be fully anonymized. Important: If there are ethical or legal restrictions to sharing your data publicly, please explain these restrictions in detail. Please see our guidelines for more information on what we consider unacceptable restrictions to publicly sharing data: http://journals.plos.org/plosone/s/data-availability#loc-unacceptable-data-access-restrictions. Note that it is not acceptable for the authors to be the sole named individuals responsible for ensuring data access.

Author’s response/explanation: The manuscript was edited to more clearly describe the minimal data set underlying the results described in the ‘Data Availability Statement’ (see p. 13, lines 304-305).

Academic editor’s comment: We note that you have stated that you will provide repository information for your data at acceptance. Should your manuscript be accepted for publication, we will hold it until you provide the relevant accession numbers or DOIs necessary to access your data. If you wish to make changes to your Data Availability statement, please describe these changes in your cover letter and we will update your Data Availability statement to reflect the information you provide.

Author’s response/explanation: We thought that this had been done mistakenly during the submission process. We would like to publish our manuscript without holding for the provision of accession or DOI. Because we do not have any repository data at the acceptance. Information concerning this has been updated in the ‘Data Availability Statement’ section.

Reviewer 1 comment: You have clearly identified the rationale for the study. Developing an education package and delivering it is an ambitious project. I wish you well with this.

Author’s response/explanation: Thank you so much for your positive comments.

Reviewer 1 comment: As a new program, I would think determining its acceptability and identifying any areas for modification are important outcomes. Yet I see no mention of acceptability/ adherence measures as outcomes.

Author’s response/explanation: The manuscript was amended with additional detail addressing this comment in the ‘Strengths and limitations’ section (see p. 9, line 228-231).

Reviewer 1 comment: How will the program’s acceptability with those delivering it and those reviewing it be assessed?

Author’s response/explanation: The comments addressed in the revised manuscript (see p. 9, lines 228-232).

Reviewer 1 comment: Will you measure adherence? If so, how? Meeting with the CHWs each month would be one measure.

Author’s response/explanation: The data collectors record completion of each of the six sessions by each of the participants. This will enable reporting on adherence to the intervention (see 9. 226-228). 

Reviewer 1 comment: Will you seek feedback on the training book?

Author’s response/explanation: Feedback from the participants was taken into consideration. Moreover, during the intervention period, there will be a fortnightly 5 to 10-minute follow-up phone call with participants to receive feedback about the education sessions and provide opportunistic counselling on the specific topics (see p. 10, line 231-232). 

Reviewer 1 comment: How do you determine compliance with the recommendations for exercise or diet? Or is this not possible?

Author’s response/explanation: The comment is addressed in the revised manuscript under the ‘strengths and limitations’ subheading (see p. 12, lines 278-282).

Reviewer 1 comment: Please include the name of the instruments used to measure the primary outcome (frailty status) and the instruments used for the secondary measures. Table 3 lists and outlines the outcome measures (i.e. the score range, number of questions and a reference for the instrument). Maybe refer to this table earlier in the text in the methods, rather than later (in the data collection), or move text from this table into the methods section.

Author’s response/explanation: The names of the instruments used to measure the primary outcome (frailty status) has been included in the revised manuscript (see p.5, line 105). The names of the instruments used to measure the secondary outcomes are also included in the revised manuscript (see p. 5, lines 108 to 112)

We agree with the comments to refer to Table 3 earlier. To make the tables in their order, Table 3 in the previous manuscript has now been amended as Table 2 in the revised manuscript (see p. 5, line 112).

Reviewer 1 comment: Line 123: as your sample calculation was an a priori analysis, change from future tense to past tense, i.e. sample size WILL be calculated, to sample size WAS calculated

Author’s response/explanation:Amendment made to sample size calculation (see p.6, line 131).

Reviewer 1 comment: Line 154 – change singular to plural, i.e. study participant’s to study participants’ involvement.

Author’s response/explanation: On the ‘Authors access to study participant information and confidentiality’ amendments from study participant’s to study participants’ (see p. 7, line 163.

Reviewer 2 comment: Abstract: Grammatical error: Statistical

Author’s response/explanation: We could not find the typo for ‘statistical’. Please provide further detail so we can address this revision.

Reviewer 2 comment: Line 126: Indicate the statistical testing method which achieves 95% power.

Author’s response/explanation: Regarding the statistical testing achieving 95%, a revision made (see p. 6, lines 133 to 134).

Reviewer 2 comment: Line 220: Categorical data is typically summarized using both frequencies and percentages.

Author’s response/explanation: Regarding categorical data, a revision made (see p 10, line 238).

Reviewer 2 comment: Line 220: State the statistical approach that will be used to test for normality of the data.

Author’s response/explanation: Regarding the statistical approach to test the normality of the data revision made (see p. 10 line 239).

Reviewer 2 comment: Since data will be collected at 3 time points, specifically indicate which outcomes and time points will be compared using Fisher’s exact tests, paired sample t-tests, and GLM with repeated measures ANVOVAs.

Author’s response/explanation: Regarding both the primary and secondary outcomes measured at three points in times, revisions made. (see p. 10, lines 240-242).

Thank you!

---

## [Decision Letter · Decision Letter 1]

21 Nov 2023

PONE-D-23-00901R1Measuring the Effects of a Nurse-led Intervention on Frailty Status of Older People Living in the Community in Ethiopia: A Protocol for a Quasi-experimental StudyPLOS ONE

Dear Dr. Kasa,

Thank you for submitting your manuscript to PLOS ONE. After careful consideration, we feel that it has merit but does not fully meet PLOS ONE’s publication criteria as it currently stands. Therefore, we invite you to submit a revised version of the manuscript that addresses the points raised during the review process.

We look forward to receiving your revised manuscript.

Kind regards,

Azmeraw Ambachew Kebede, MSc

Academic Editor

PLOS ONE

Journal Requirements:

Reviewers' comments:

Reviewer's Responses to Questions

**Comments to the Author**

1. Does the manuscript provide a valid rationale for the proposed study, with clearly identified and justified research questions?

Reviewer #3: Yes

Reviewer #4: Yes

2. Is the protocol technically sound and planned in a manner that will lead to a meaningful outcome and allow testing the stated hypotheses?

Reviewer #3: Yes

Reviewer #4: Yes

3. Is the methodology feasible and described in sufficient detail to allow the work to be replicable?

Reviewer #3: Yes

Reviewer #4: No

4. Have the authors described where all data underlying the findings will be made available when the study is complete?

Reviewer #3: Yes

Reviewer #4: Yes

5. Is the manuscript presented in an intelligible fashion and written in standard English?

Reviewer #3: Yes

Reviewer #4: Yes

6. Review Comments to the Author

You may also provide optional suggestions and comments to authors that they might find helpful in planning their study.

Reviewer #3: The authors have responded well to comments of the reviewers. The manuscript is interesting and well-written.

Regarding reviewer 1, a simple measure of compliance could be the percentage of participants completing all 6 sessions as a secondary endpoint.

Minor comments:

1. Abstract: please specify the expected number of participants

2. Methods: I recommendend to add the number of falls among the secondary outcomes "number of falls in the 3 months prior to intervention, number of falls at 3 months, 6 months and 12 months").

3. Methods: non-inclusion criteria, replace "cognitive impairment" by "major cognitive impairment"

4. Methods: Student's t-test will be probably used to compare the level of frailty with categorical variables (sex, education etc). Add Student's t-test.

I recommend this manuscript for publication.

Best regards

Boucaud-Maitre Denis

Reviewer #4: 1.Line 111.6, it is unclear why the study was registered PROSPECTIVELY 3 months after data collection. Should it be RETROSPECTIVELY as registration of the study happened after the start of data collection?

2.Line 113-133, If the data is assumed to be normally distributed, why paired-t test is not used for sample size calculation? Also, please clarify the end point of the primary outcome for sample size calculation, either from T0 to T1 or T0 to T2. The author should state clearly which outcome measure is used for sample size calculation as the assumed effect size of 0.5 means it is assumed that the effect size of the intervention on the primary outcome of the study is 0.5, which should also be supported by evidence.

3.Line 137-144, it is unclear how the subjects will be recruited, either from the poster or CHW proactively to recruit subjects through the available list. More specifically, which sampling method will be used? The section Recruitment should place after the section on Eligibility.

4.Line 169, for the NLI program, the sessions can be distinct and interconnected (independent means they are unconnected)

5.Line 226. Some descriptions of the outcome measures should be provided in the text, rather than provided in the hypothesis under Study design and hypothesis. Moreover, some of the outcomes were not touched such as recent appetite. Acceptability of the program should also be measured from the perspectives of the participants on top of the interventionist (i.e., CHWs)

6.Line 272-285, Please group the points for two subsections of strengths and limitations of the study.

7. PLOS authors have the option to publish the peer review history of their article (what does this mean?). If published, this will include your full peer review and any attached files.

Reviewer #3: **Yes: **Boucaud-Maitre Denis

Reviewer #4: No

---

## [Author Response · Author response to Decision Letter 1]

1 Dec 2023

Reviewer 3 Comments: Abstract: please specify the expected number of participants

Authors' explanation: Expected number of participants specified in abstract (see p. 2, line 38).

Reviewer 3 Comments: Methods: I recommendend to add the number of falls among the secondary outcomes "number of falls in the 3 months prior to intervention, number of falls at 3 months, 6 months, and 12 months").

Authors' explanation: Thank you for your genuine comments on this matter. To be clear the frequency of the data collection, will be immediately before the intervention (baseline) (T0), immediately post-intervention (T1), and at 12 weeks post-intervention (T2). Considering the literature and our study aim the number of falls is included in the baseline data collection. The falls history and frequency of falls of study participants in the last 12 months will be recorded at baseline. The details of our primary and secondary outcomes are to be measured at three different points in time (see p.10 lines 102 to 112).

Reviewer 3 Comments: Methods: non-inclusion criteria, replace "cognitive impairment" by "major cognitive impairment"

Authors' explanation: The term "cognitive impairment" is replaced by "major cognitive impairment" (see p. 16, line 149).

Reviewer 3 Comments: Methods: Student's t-test will be probably used to compare the level of frailty with categorical variables (sex, education, etc). Add Student's t-test.

Authors' explanation: Student's t-test will used to compare the level of frailty with categorical variables (see p. 24, line 274-275).

Reviewer 4 Comments: Line 111.6, it is unclear why the study was registered PROSPECTIVELY 3 months after data collection. Should it be RETROSPECTIVELY as registration of the study happened after the start of data collection?

Authors' explanation: Thank you for your observation on this matter. This has been addressed in the revised manuscript (see p. 15, line 121).

Reviewer 4 Comments: Line 113-133, If the data is assumed to be normally distributed, why paired-t test is not used for sample size calculation? Also, please clarify the end point of the primary outcome for sample size calculation, either from T0 to T1 or T0 to T2. 

Authors' explanation: Even though we used the Wilcoxon signed rank test we considered normal parent distribution methods to determine the sample using G-Power. The endpoint for the primary outcome for sample size calculation is from T0 to T2.

Reviewer 4 Comments: The author should state clearly which outcome measure is used for sample size calculation as the assumed effect size of 0.5 means it is assumed that the effect size of the intervention on the primary outcome of the study is 0.5, which should also be supported by evidence.

Authors' explanation: The primary outcome measure, frailty, was used for sample size calculation. 

We set the effect size at 0.5 as it is often used as a practical rule of thumb, especially when there is no prior knowledge or specific expectations about the effect size. It provides a balance between detecting meaningful effects and ensuring a feasible sample size. A moderate effect size of 0.5 is considered large enough to be practically significant in many fields (Althubaiti, 2023).

Reviewer 4 Comments: Line 137-144, it is unclear how the subjects will be recruited, either from the poster or CHW proactively to recruit subjects through the available list. More specifically, which sampling method will be used? 

Authors' explanation: The reason for distributing a poster containing the aim of the study, eligibility criteria, and benefits of participating in the study was to make it readily available to anyone who need to take part in the study. However, we have made a revision so that it will be clearer how the subjects will be recruited. The study participants will be recruited using convenient sampling (see p. 16, lines 152 to 153, lines 155 to 156).

Reviewer 4 Comments: The section Recruitment should place after the section on Eligibility.

Authors' explanation: A revision has been made based on the comment (see p. 16 line 142 and 152).

Reviewer 4 Comments: Line 169, for the NLI program, the sessions can be distinct and interconnected (independent means they are unconnected)

Authors' explanation: A revision has been made based on the comment (see p. 17 line 172).

Reviewer 4 Comments: Line 226. Some descriptions of the outcome measures should be provided in the text, rather than provided in the hypothesis under Study design and hypothesis. Moreover, some of the outcomes were not touched such as recent appetite. 

Authors' explanation: Thank you for your insights on this section. 

In the study design and hypothesis section, we tried to amend the flow. 

We believe that the outcome measurement descriptions are provided and explained at the end of the hypothesis and design section. Moreover, a brief description of each outcome measure, measurement tools and frequency of measurement has been indicated in Table 2. ‘Recent appetite’ has been included as one of the outcome measures in the revised manuscript (see p. 10, lines 109). The measurement tool description for this outcome has been described in Table 2, section VI.

Reviewer 4 Comments: Acceptability of the program should also be measured from the perspectives of the participants on top of the interventionist (i.e., CHWs)

Authors' explanation: Acceptability of the program from the perspective of the participants will be recorded. This has been stated in the revised manuscript (see p. 23 lines 263 to 264). 

Reviewer 4 Comments: Line 272-285, Please group the points for two subsections of strengths and limitations of the study.

Authors' explanation: Two subsections have been made in the revised version (see p. 25 line 304 and line 311).

---

## [Editor Report · Decision Letter 2]

8 Dec 2023

Measuring the Effects of a Nurse-led Intervention on Frailty Status of Older People Living in the Community in Ethiopia: A Protocol for a Quasi-experimental Study

PONE-D-23-00901R2

Dear Dr. Kasa,

We’re pleased to inform you that your manuscript has been judged scientifically suitable for publication and will be formally accepted for publication once it meets all outstanding technical requirements.

Kind regards,

Azmeraw Ambachew Kebede, MSc

Academic Editor

PLOS ONE
---

## [Editor Report · Acceptance letter]

8 Jan 2024

PONE-D-23-00901R2 

PLOS ONE

Dear Dr. Kasa, 

I'm pleased to inform you that your manuscript has been deemed suitable for publication in PLOS ONE. Congratulations! Your manuscript is now being handed over to our production team.

Kind regards, 

on behalf of

Mr. Azmeraw Ambachew Kebede 

Academic Editor

PLOS ONE